# Influence of Evacuation Policy on Clearance Time under Large-Scale Chemical Accident: An Agent-Based Modeling

**DOI:** 10.3390/ijerph17249442

**Published:** 2020-12-16

**Authors:** Minjun Kim, Gi-Hyoug Cho

**Affiliations:** School of Urban and Environmental Engineering, Ulsan National Institute of Science and Technology, 50 UNIST-gil, Ulju-gun, Ulsan 44919, Korea; min2412@unist.ac.kr

**Keywords:** urban resilience, evacuation model, TRANSIMS, ALOHA, chemical accident, agent-based model

## Abstract

Large-scale chemical accidents that occur near areas with large populations can cause significant damage not only to employees in a workplace but also to residents near the accident site. Despite the increasing frequency and severity of chemical accidents, few researchers have argued for the necessity of developing scenarios and simulation models for these accidents. Combining the TRANSIMS (Transportation Analysis and Simulation System) agent-based model with the ALOHA (Areal Location of Hazardous Atmospheres) dispersion model, this study aims to develop a modeling framework for simulating emergency evacuations in response to large-scale chemical accidents. The baseline accident scenario assumed the simultaneous leakage of toxic chemicals from industrial complexes near residential areas. The ALOHA model results showed that approximately 60% of residents in the scenario’s city were required to evacuate their homes. The majority of evacuees completed their evacuations within 5 h in the baseline scenario (evacuating maximum number of private vehicles without any intervention), while the distribution of the population and street network density caused geographical variability in clearance time. Clearance time can be significantly reduced by changing both the evacuees’ behaviors and the evacuation policy, which suggests the necessity for proper public intervention when the mass evacuation of residents is required due to chemical accidents.

## 1. Introduction

During the last decades, natural and man-made disasters in urban areas have consistently put large populations at risk. Hence, alleviating damage and protecting lives through an effective evacuation plan have become essential roles for emergency managers [1]. However, evacuating large numbers of people in a timely manner is not always practically feasible, as populations often tend to panic when faced with disasters and, thus, have difficulty making reasonable decisions (e.g., evacuation departure time, transportation mode, and evacuation shelter choice) by themselves [2]. Moreover, severe traffic congestion during evacuations can make it challenging for evacuees to arrive at the appropriate shelters at the right time [3]. In such complex and uncertain situations, local governments and emergency managers are often required to prepare systematic evacuation plans in advance.

To this end, previous studies have proposed a number of methodologies to model emergency evacuation in the disaster-affected area. Several physical models, including the Social Force Model [4,5,6] and Cellular Automata (CA) Model [7,8], have been proposed to simulate crowd evacuation. One of the common characteristics of those approaches is that evacuees are modeled to act based on the behavioral rules researchers predefined. To establish reasonable behavioral rules by examining various factors that affect evacuees’ behavioral patterns, several statistical analysis techniques, such as logit models, have been utilized [9,10]. More recently, advanced analyzing strategies, including machine learning [11,12,13] and driving simulation [14,15], were actively utilized for modeling evacuation behavior. These studies have identified the hidden behavioral rules of evacuation by utilizing computer science and simulators.

Agent-based evacuation modeling has been highlighted as one of the most effective tools for developing evacuation strategies due to its capacity to model multiple evacuation scenarios at relatively low cost [16]. Using agent-based systems for evacuation modeling is preferable to traditional static modeling tools for several reasons. First, agent-based systems use a bottom-up approach in which system control is governed by autonomous agents’ behavior [17]. This bottom-up approach makes it easier for practitioners to develop multiple evacuation scenarios by adjusting several local parameters. Second, the efficiency of multiple evacuation scenarios can be evaluated at the agent level; by contrast, only the aggregated efficiency of an evacuation can be analyzed in traditional static models [18]. Evaluating multiple evacuation scenarios provides a better understanding of evacuees’ behavioral characteristics, which aids in developing specific evacuation strategies for local government and emergency managers.

For these reasons, a significant amount of literature has adopted agent-based methodologies to simulate the evacuation of large populations during natural hazards, such as hurricanes [19,20], earthquakes [21,22], and floods [23,24], or technological accidents, such as fire [25,26] and civil violence [27]. Those studies have shown that agent-based modeling can be successfully utilized to simulate the evacuation of multiple evacuees in various disaster scenarios. In particular, there has been an increase in the number of studies that conducted agent-based evacuation modeling in the context of chemical accidents [28,29,30,31]. In those studies, the two modeling approaches are generally integrated: (1) dispersion modeling of accidental gas leakage and (2) evacuation modeling of populations within the affected area. While previous studies have developed modeling techniques to simulate mass evacuation after a chemical accident, most of them assumed that the leaked toxic chemicals disperse within industrial areas, where evacuating large populations is not required. It seems mainly due to the low frequency of large-scale chemical accidents. Further, a mass evacuation for large-scale chemical accidents is difficult to be modeled since doing so requires site-specific databases, including demographics and physical conditions in a given area.

However, as we learned from the Seveso disaster experience in 1976 or Bhopal in 1984, large-scale chemical accidents that occur near high-population areas can cause significant damage not only to employees in a workplace but also to residents near the accident site [32]. To reduce damage and casualties resulting from large-scale chemical accidents, it is necessary to evaluate the performance of large-scale emergency evacuations in local contexts. This study presents an evacuation modeling framework for large-scale chemical accidents by combining two individual modeling tools: (1) Areal Location of Hazardous Atmospheres (ALOHA) and (2) Transportation Analysis and Simulation System (TRANSIMS). The first model was used to estimate the spatial extent of a chemical release. Local databases were used as input data for the estimation of the spatial extent of toxic chemicals release and the number of evacuation demands. Then, we used the second model to simulate the mass evacuation of the estimated number of evacuees within the affected area. To reduce the traffic congestion that could occur during the evacuation, we selected three evacuation staging sites and designated evacuees based on their distance from them. Staging sites were chosen as three districts outside of the toxicity threat zone. To evaluate the performance of the evacuation scenario, clearance time was used as an indicator. Finally, we conducted a sensitivity analysis of the evacuation’s performance by modifying several crucial parameters, such as the proportion of private vehicles used for the evacuation, the time range for the evacuation departure, and the number of vehicle lanes available. The final section of this paper summarizes the findings of our research and discusses future work.

## 2. Materials and Methods

### 2.1. Study Area and Baseline Scenario

Ulsan is the seventh-largest metropolitan area in the southeastern part of Korea, encompassing an area of 1060 km^2^ with a population of 1.17 million [33]. Two national industrial complexes are located near the city along its coastline—Mipo and Onsan national industrial complexes. According to the Ministry of Environment, more than 50 million tons of toxic chemicals—including formaldehyde, toluene, hydrogen fluoride, and hydrogen sulfide—are annually used in the complexes accounting for approximately 30 percent of total national usage. The National Institute of Chemical Safety (NICS) states that factories storing more than the minimum storage volume should submit risk management plans for chemical accidents. Within the two national industrial complexes, 51 chemical plants that are currently being operated submitted risk management plans (Figure 1).

The study area is understood to be one of the most vulnerable regions to disastrous chemical accidents for two reasons. First, since the two industrial complexes are over 40 years old, the deterioration of chemical pipelines and storage facilities has progressed considerably. Accidental failure of chemical storage units can leak a voluminous amount of toxic chemicals. Second, residential districts are located near the industrial complexes: a population of 0.94 million (81% of Ulsan’s total population) resides within a 10 km radius of the complexes. Toxic chemicals released from the complexes could spread rapidly toward the residential area, putting large numbers of people at tremendous risk.

To evaluate mass evacuation performance in the study area, we developed the baseline scenario for a large-scale chemical accident and mass evacuation (Figure 2). In the baseline scenario, we assumed that toxic chemicals stored in 51 chemical plants within the complex were simultaneously released at 2 a.m. on a weekday, when most residents are likely to be in their own homes [34]. Under very stable weather conditions, released toxic chemicals spread rapidly toward the nearby heavily populated area, and all households within the affected area were instructed to evacuate from their homes to the closest staging site. All evacuees were assumed to start an evacuation within 2 h after the occurrence of a chemical accident. Households with their own private vehicles were designated to use them for the evacuation; otherwise, forty-seater buses for the public were used by the local government.

### 2.2. Modeling Framework

To simulate a baseline evacuation scenario for a disastrous chemical accident, we integrated two individual modeling tools into one framework: (1) the Areal Location of Hazardous Atmospheres (ALOHA) model and (2) the Transportation Analysis and Simulation System (TRANSIMS) model (Figure 3). The first model is used for estimating the spatial extent of a toxic chemical release and the related evacuation demands. In this model, several site-specific parameters (e.g., type of toxic chemicals, weather conditions) and demographic information (e.g., number of households, the proportion of vehicle ownership) were defined using relevant local databases. The second model was utilized to simulate residents’ mass evacuation based on estimated evacuation demands projected in the first model. This model comprises four modules: a network converter, a trip generator, a route planner, and a microsimulator. The final output of the model is evacuees’ clearance time, which indicates the time required to evacuate people from their origin to the emergency staging sites.

### 2.3. Estimating Spatial Extent of Chemical Accident

To estimate the projected spatial extent of a chemical release, we employed the Areal Location of Hazardous Atmospheres (ALOHA) model, a stand-alone simulation tool developed by the National Oceanic Atmospheric Administration (NOAA). ALOHA, which is based upon the Gaussian dispersion model of continuous, buoyant air pollution flumes [35], can simulate the accidental release of over 900 chemicals. Since the model is freeware and easily compatible with other geographic tools, such as Google Earth and ArcGIS, previous studies have used the ALOHA model to simulate hazardous substance release [36,37,38,39]. We employed version 5.4.7 (released in September 2016) of the software, which is the latest version.

#### 2.3.1. ALOHA Input Data

The data input into the ALOHA model comprises three parts: local atmospheric conditions, the identity of chemicals, and details about the spill scenario [40]. To simulate the worst-case scenario for a chemical release, this study followed guidelines suggested by NICS, developed for chemical companies to establish risk management plans. Table 1 summarizes the ALOHA input data used in our research.

To define the chemicals in a scenario, ALOHA users need to choose the type of released chemicals and whether certain chemicals are released as “pure chemicals” or as “solutions”. This study assumed that the pure forms of the 19 toxic chemicals used in the chemical complexes were released. There are four source types for the chemical releases in the model: direct, puddle, tank, and gas pipeline. One of the main differences among those source types is leakage rate consistency. For this study, we chose to use the direct source type of leakage as our model input due to our lack of information about the storage units’ physical structure at the sites. To determine the leakage rate of a given chemical, we used the minimum storage volume of chemicals designated by NICS. Chemicals were set to be released consistently from the ground-level storage for 60 min after the accident.

#### 2.3.2. Identification of Toxicity Threat Zone

After setting all the necessary input data, users need to choose one or more toxic Levels of Concern (LOCs) to identify the chemical release threat zones. A toxic LOC refers to the threshold concentrations of exposure that can be a threat to human health. Several toxic LOC guidelines available in ALOHA are: Acute Exposure Guideline Levels (AEGLs), Emergency Response Planning Guidelines (ERPGs), Protective Action Criteria (PAC), and Immediately Dangerous to Life or Health (IDLH). We selected ERPGs as the most suitable LOC for the study since it estimates wider threat zones than the others. A number of recent studies have utilized ERPGs as exposure guidelines for identifying the toxicity of threat zones [41,42,43,44].

ERPGs have three tiers of exposure values for each chemical. The first tier (ERPG-1) is a mild effects threshold; the second tier (ERPG-2) is an escape-impairment threshold; and the third tier (ERPG-3) is a life-threatening effects threshold. Using ALOHA, we simulated the ERPG threat zones of 19 toxic chemicals. Figure 4 shows the identified ERPG threat zones provided by the model. The threat zones are represented as yellow, orange, and red zones, corresponding with ERPG-1, ERPG-2, and ERPG-3, respectively. The dotted line shows wind direction confidence, which indicates the possible extent of the chemical release as wind direction changes. Since our accident scenario used stable weather conditions without any prevailing wind direction, the radius shown by the dotted circle was adopted as the spatial extent of the chemical release. Each chemical released will have its unique ERPG threat zones.

### 2.4. Agent-Based Evacuation Modeling

Based on the chemical release’s identified spatial extent, we developed an agent-based model for mass evacuation using the Transportation Analysis and Simulation System (TRANSIMS). TRANSIMS is an agent-based model that simulates individual travelers’ behavior and their multimodal transportation options using a cellular automata microsimulator. The model, developed by the Los Alamos National Laboratory, is currently available as open-source software (version 4.0.8). One of the strengths of TRANSIMS is that it can be easily modified to the user’s purpose. The software was originally designed to evaluate transportation systems, but previous studies have demonstrated that TRANSIMS is an effective and reliable software for evacuation modeling [45,46,47].

#### 2.4.1. Network Converter

In a network converter, users need to provide input datasets so that the module can convert them into a TRANSIMS network system. Node, link, and zone files are the three key input network files. Table 2 lists the information required for each input file. To create network files for the study area, we utilized network datasets provided by the Korea Transport Database (KTDB).

In the node file, the node ID and the x,y coordinates of each node should be included. We entered the ID and coordinates of 7021 intersection points within the study area. Using a defined node, we constructed 10,031 links for the input link file. For each link, the number, length, type, and usage of lanes were assigned based on KTDB. A zone file is not a mandatory component in a network converter, but this study employed it to define evacuation departure and arrival locations. We defined the ID and coordinates of 56 administrative districts within the study area as zone files.

#### 2.4.2. Trip Generator

A trip generator is a module that estimates travel demands. Two important input files are required for this step: origin–destination (O-D) and diurnal distribution tables. First, O-D tables contain data on the number of trips between two zones. We assumed that all residents within the chemical release’s spatial extent evacuated from their residential districts to the closest emergency staging sites. Three administrative districts outside of the toxicity threat zone, but close enough to it, were chosen as the emergency staging sites’ locations. Second, the number of trips was determined by the number of private and public vehicles available within the threat zone. Among households within the study site, 76.2% have their private vehicle, and they were set as using a vehicle for evacuation. Households that do not own vehicles were assumed to use forty-seater buses for the evacuation. Finally, the residents’ evacuation departures were set to occur consistently within 2 h after the accidents occurred.

#### 2.4.3. Route Planner and Microsimulator

Based on the network system and trips generated in the preceding modules, the route planner and the microsimulator simulate the vehicles’ actual travel behavior. In the route planner, TRANSIMS computes the shortest paths for each vehicle in the network system without considering any interactions. Once the plans are generated for all vehicles, they are implemented in the microsimulator. In the microsimulator module, each vehicle follows a designated route plan based on a cellular automata model. Cellular automata refer to a cell-based system wherein an agent in a certain cell can move to another cell according to a set of rules [48]. The size of each cell and the agent’s speed are critical data in a cellular automata model. We used a TRANSIMS default cell length of 7.5 m and a maximum speed of 3 cells per second (approximately 80 km/h). By calculating the interactions among agents, a microsimulator provides the actual travel routes and travel times of evacuees.

### 2.5. Sensitivity Testing

One of the clear indicators of evacuation efficiency is clearance time, which refers to the time required to complete all evacuees’ trips to designated emergency staging sites [49]. Clearance time may depend on various factors, including travel demands, background traffic, traffic control policies, evacuation route assignment, and response rate [50]. This study used four factors to test the sensitivity of clearance times: the proportion of private vehicles, departure time interval, phase control, and lane policies.

The proportion of private vehicles determines evacuation transportation demands. This study tested clearance times when 0, 25, 50, 75, and 100% of the maximum private vehicles were used. Since we assumed that one household has four members, one public forty-seater bus replaced 10 private vehicles. Therefore, when the number of private vehicles decreased by 25%, that of buses increased by 2.5%. Second, we varied the range of evacuees’ departure times by 1 h from 1 h to 4 h to test the departure time intervals’ sensitivity to evacuation efficiency. Compared to evacuation departures within 4 h, the number of vehicles generated every 15 min increased by four times in a 1 h evacuation. The number of vehicles generated was equally distributed every 15 min. For evacuation phase control, two different phasing policies were tested: normal and phased. A constant number of evacuation demands were generated every 15 min regardless of the designated emergency staging sites in the normal case. In the phased case, administrative districts were divided into three groups based on their distance from the accident site, and the households located closest to the site were evacuated earlier than the others. Lastly, we tested the effect of lane policies, including contraflow and reduced lanes, on clearance time. In contraflow, the number of available lanes increased as those lanes running in the opposite direction of the sites were allowed to be used for the evacuation. In reduced lanes, the number of available lanes for the evacuation was reduced by one because one lane was restricted for emergency vehicles.

## 3. Results

### 3.1. Spatial Extent of Toxic Chemical Release

We first estimated the spatial extent of a toxic chemical release at each ERPG level using the ALOHA model (Table 3). For ERPG-2, formaldehyde (7.84 km), hydrogen chloride (7.7 km), nitric acid (6.05 km), sulfuric acid (5.32 km), and chlorine (5 km) had substantially larger spatial extents of chemical release compared to other chemicals. Based on this estimation, we identified the spatial extent of the chemical release of 51 chemical plants within the complexes. This result was used to determine the locations of the evacuees and the evacuation demand, which is the first step in modeling an evacuation. When more than one toxic chemical was stored in a chemical plant, the one with the largest radius was used for the estimation. By merging the spatial extents of chemical releases from 51 chemical plants, we defined the spatial extent of an evacuation at the ERPG-2 and ERPG-3 levels (Figure 5).

The areas of the ERPG-2 and ERPG-3 level threat zones were 479.83 km^2^ and 241.90 km^2^, respectively. Besides, the number of households within each zone was estimated at between 283,920 and 239,221. Among the 56 administration districts in the study area, 46 and 33 administration districts were partially or fully included in each ERPG threat zone, respectively. The total number of vehicles (agents) was 223,083 at the ERPG-2 level and 187,962 at the ERPG-3 level in the baseline scenario. For the ERPG-2 level, the number of vehicles destined for emergency staging sites 1, 2, and 3 was estimated to be 72,257, 98,688, and 45,402, respectively.

### 3.2. Baseline Evacuation Modeling

In the baseline scenario (See Figure 2), 223,083 vehicles (agents) were evacuated from a home to a designated staging site within 2 h of the occurrence of the chemical accident. We analyzed the temporal and spatial patterns of evacuees based on the outcomes of the TRANSIMS model, including their average speed, clearance time, and clearance rate trends. Figure 6 and Table 4 represent the average speed of vehicles (agents) for the evacuation from 1 to 6 h after the occurrence of the chemical accident. Among 6185 road segments constructed for this model, almost every segment (99.35%) was used for the evacuation more than once. Up until 1 h from the accident’s occurrence, the average speed of the vehicles was higher than 20 km/h in the majority (67.58%) of road segments, except for some segments near the city center. A large population resides near the city center, resulting in traffic congestion (less than 10 km/h) in the early stage of evacuation. The road segments near the city center experienced more severe traffic congestion up to 3 h from the accident occurrence and the congestion expanded to major roads, where traffic flows to reach the designated staging site joined. Segments with speeds of less than 10 km/h dramatically increased from 9 to 25% between 1 and 3 h after the accident. At 4 h, severe traffic congestion for road segments in the city center was largely relieved and congestion mainly remained on the major roads (9.22%). Since all evacuees started an evacuation within 120 min, most had joined the major roads toward the emergency staging sites. The evacuation process was close to an end in 6 h. In order to evaluate the evacuation performance of each administration district, times required for the evacuation of 25%, 50%, 75%, and 90% of the evacuees were estimated in the baseline scenario (Figure 7). Districts assigned to emergency staging sites 1, 2, and 3 are shown with a blue, red, and green line, respectively.

This analysis allows us to understand geographical variations in evacuation performance and to identify districts that are most vulnerable in chemical accidents. For the clearance of the first 25% of evacuees, administration districts located far from the staging site generally required longer evacuation times, but there was no substantial difference in clearance times across districts, which ranged from 1 h 5 min to 1 h 39 min. Due to geographical variations in population and network density, some districts took longer to reach staging site 1 even though they were geographically closer to the destination. For instance, districts with large populations and dense street networks tended to require more time to join major roads. For the clearance of the first 50% of evacuees, the difference between the shortest (1 h 52 min) and the longest (4 h 11 min) clearance time became much larger. These results suggest that substantial delays resulting from traffic congestion began to occur at this stage. In particular, half of evacuees from the district adjacent to the East Sea could not reach the designated staging site until four hours after the accident. Congestion and bottlenecks worsened to such an extent that longer than four hours was required for the evacuation of 75% of households in most of the districts within the study area. Severe traffic congestion began to be relived only after four hours, but clearance time for 90% of households ranged from 4 h 44 min to 6 h 38 min We observed that some agents in the simulation model could not find the shortest path to a designated staging site or became stuck somewhere. Due to these unexpected behaviors of agents, we presumed that the estimated clearance time for more than 90% of households might be unreliable.

Figure 8 shows the temporal trend in clearance rates for each staging site. Regardless of the locations of the emergency staging sites, the time required for a complete evacuation was approximately 6 h after the evacuation process began. The clearance rate linearly increased to 45% 2 h after the outbreak of the chemical accident. During this period, the clearance rate depended significantly on the geographical distance of each district from the designated emergency staging sites. After 2 h, however, the increasing rate of clearance quickly slowed down for all three emergency staging sites due to severe traffic congestion in the network, which corresponds with the results shown in Figure 6. Such congestion continued for approximately 1 h 30 min. until clearance rate reached 50%. After 3 h 30 min, clearance rates started to sharply increase again as traffic congestion in most of the internal networks was relieved. The clearance rate of staging site 2 reached 90% in a slightly shorter time than the others, but, in general, the temporal patterns of evacuation were consistent across the three staging site locations.

### 3.3. Sensitivity Analysis

Based on the results from the baseline evacuation modeling scenario, we tested the sensitivity of network clearance time to assumptions made for four factors (Figure 9 and Table 5). The first factor is the proportion of private vehicles used for evacuation. By varying the proportions of private vehicle travel from 0% (no private vehicle) to 100% (maximum number of private vehicles), the total number of vehicles was estimated at between 22,308 and 223,083. Not surprisingly, the results showed that the clearance time required for evacuation generally increases as the proportion of private vehicles increases. The increasing clearance rate trend, however, varied by the proportion of private vehicle use. In 50%, 75%, and 100% of scenarios, traffic congestion occurred from 2 to 3 h after the evacuation process began, while such congestion was not found in 0% and 25% scenarios. When more than half of the evacuees chose private vehicles, the overall traffic demand exceeded the current capacity of the road network in the study area and evacuation performance substantially reduced.

Sensitivity of clearance time to departure time was tested by varying it from 1 to 4 h by 1 h increments. Compared to the baseline (2 h range) scenario, we found that as the range of evacuation departure time increased, the time required to relieve traffic congestion was decreased. Compared to the results in the baseline scenario, the clearance rate for the 1 h range reached 70% more quickly, within 2 h. However, the increase of the clearance rate was largely stagnant after 2 h due to traffic congestion. In contrast to the results of the baseline scenario, the congestion in the 1 h range was not relieved until the end of the evacuation process so that the clearance rate of the baseline scenario (the 2 h range) became higher than that of the 1 h range departure after 4.5 h.

We did not find a significant improvement in clearance rates when phased evacuations were implemented. One possible explanation for this result is that phasing policies cannot affect the total number of vehicles in the evacuation and, thus, do not affect overall clearance rates. Although the departure timing of evacuees is affected by phased evacuation, the influence of the policy was not as manifest as the influence of departure time range. With regard to lane policies, we found that the clearance rate was slightly increased by allowing contraflow and decreased by yielding one lane to emergency vehicles. However, in our study results, the influence on capacity increase of allowing contraflow did not substantially improve evacuation efficiency. At 4 h after evacuation, the clearance rate of the contraflow policy was 11% higher than that of the baseline, while the difference between the two was less than 5% at other time points.

## 4. Discussion

By integrating agent-based evacuation models with the ALOHA model, this study developed a modeling framework for simulating an emergency evacuation caused by a disastrous chemical accident. The effectiveness of evacuation was assessed with the modeling framework applied to Ulsan, Korea, and the results showed that approximately 60% of total residents within the study area were required to evacuate, and the majority of evacuees completed their evacuations within 6 h in the baseline scenario. This study demonstrated that severe traffic congestion started to occur within two hours of the evacuation and continued until four hours after the accidents had occurred. Level of congestion during the evacuation appears to be determined by whether the road is located near a populated city center or not. Besides, it suggests the necessity of finding the most vulnerable regions for evacuations and providing alternatives to them.

This paper also examined the sensitivity of the evacuation’s effectiveness to several key assumptions, including the proportion of private vehicles, evacuation departure time, phasing policies, and lane policies, made about evacuation behaviors and planning interventions. Based on our analysis, reducing private vehicles and changing departure time period appear to significantly improve evacuation efficiency by lessening traffic congestion among evacuees. However, an evacuation policy such as phased evacuation or lane policy had a relatively limited effect on the evacuation’s efficiency. It implies that reducing the number of vehicles on the road per hour is more effective than other physical interventions.

Findings from this study suggest several strategies to shorten evacuees’ clearance time and thus improve the efficiency of evacuation. First, alleviating traffic congestion in the urban center can be one of the key strategies to increase overall evacuation efficiency. Providing contraflow lanes and alternative routes to staging sites would be critical to practitioners in emergency evacuations. Second, regulating private vehicle use during an evacuation can reduce the congestion and thus increase evacuation efficiency. It is not surprising, but it highlights the necessity of utilizing public transit during the evacuation instead of private vehicles. Recent studies have shown that transit-based evacuation significantly reduced the clearance time [51,52]. Third, interventions for evacuation time periods can reduce overall clearance time since it disperses evacuation demands. It suggests that regional authorities and evacuation planners should establish sequential evacuation plans for the region and encourage people to be educated and trained in advance.

## 5. Conclusions

Previous evacuation models applied to the worst-case hazard scenarios assumed that all city residents were required to evacuate. Such assumptions often overestimate evacuation demands and clearance time in the evacuation models, and, thus, practitioners may have difficulties in developing efficient evacuation plans with limited resources. This study is one of the few studies that have incorporated the estimation of chemical accidents’ spatial extent into evacuation modeling. Besides, this study is originated in the emergency management field in that it suggests combined evacuation models using two individual simulation tools (ALOHA and TRANSIMS) and its usability. By doing so, the model estimates the reasonable spatial extent of disastrous chemical accidents and evacuation demands in the local context.

However, it is important to note that this is not a predictive tool that can precisely simulate evacuees’ individual behaviors; rather, its value lies in evaluating overall evacuation efficiency in a worst-case scenario chemical accident. Since such a worst-case scenario has not occurred previously, the validity of the assumptions made for the base scenario cannot be tested properly. In light of this, we propose several modeling improvements for future research.

TRANSIMS employed the shortest path assumption [53]. However, in reality, evacuees occasionally avoid traffic congestion by choosing detours [54]. The TRANSIMS visualizer showed that agents did not find alternative routes well when severe traffic congestion occurred, which may induce unnecessary wait times for vehicles at intersections, resulting in excessive clearance time. Improving interactions among different agents during the disaster evacuation scenario would likely provide more accurate simulation results.

Last, we assumed that the disaster occurred at 2 a.m. and that all evacuees originated from their homes. This assumption seems reasonable since the Korea time-use survey showed that most of the residents are at home at 2 a.m. This spatiotemporal pattern of the exposed population is one of the essential factors for understanding evacuation flows. Identifying populations’ spatial and temporal patterns with a time-use survey and analyzing dynamic exposure when the disaster occurs may significantly increase the practical usability of this modeling framework in disaster management.

Despite several technical limitations, this study is originated in the emergency management field in that it suggests the combined evacuation models with two individual simulation tools (ALOHA and TRANSIMS) and its usability. By doing so, the model estimates reasonable spatial extent of disastrous chemical accidents and evacuation demands in the local context. In addition, we evaluated the efficacy of several planning interventions to clearance time and proportion of private vehicles and evacuation departure time periods were found to be some of the suitable factors to improve efficiency of evacuation. When properly used, emergency managers and practitioners may improve reliability of local evacuation plans and, thus, reduce casualties in case disastrous chemical accidents occur.

## Figures and Tables

**Figure 1 ijerph-17-09442-f001:**
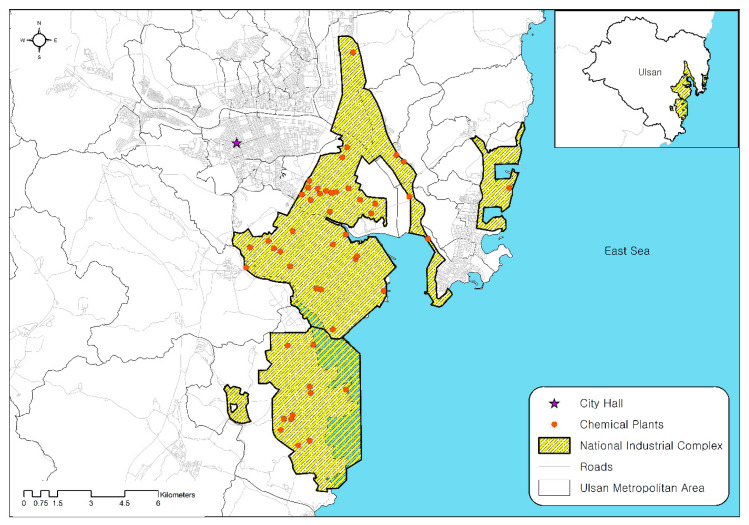
National industrial complexes and 51 chemical plants located within Ulsan.

**Figure 2 ijerph-17-09442-f002:**
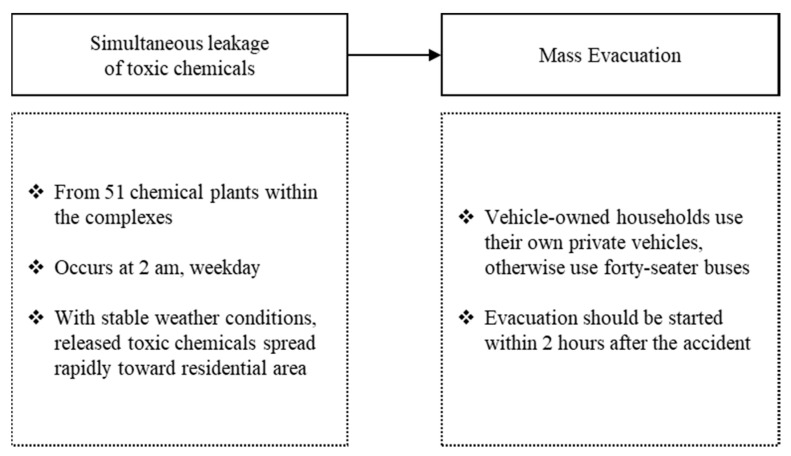
Baseline scenario of chemical accident and mass evacuation.

**Figure 3 ijerph-17-09442-f003:**
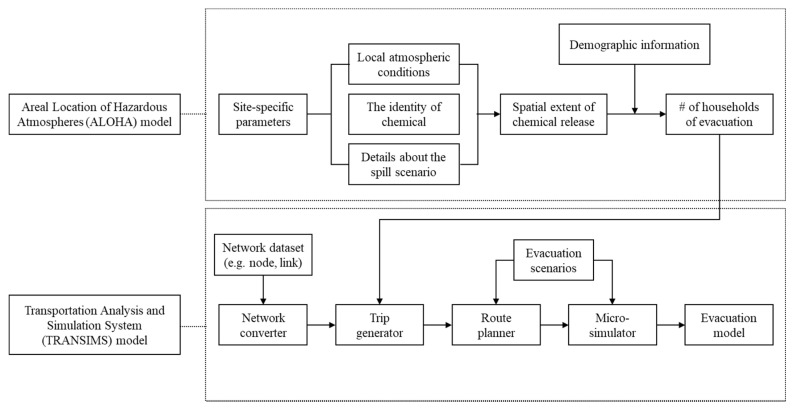
Modeling framework of the study.

**Figure 4 ijerph-17-09442-f004:**
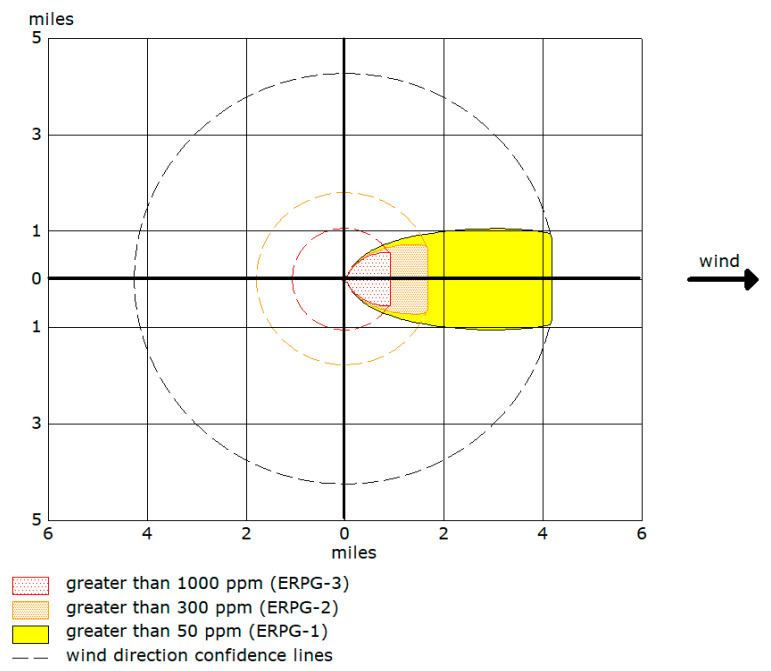
Example of identified ERPG threat zones in ALOHA.

**Figure 5 ijerph-17-09442-f005:**
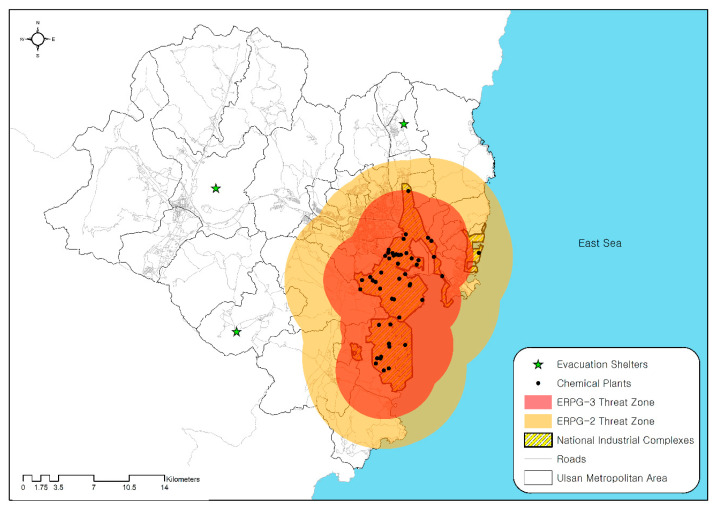
Example of identified ERPG threat zones in ALOHA.

**Figure 6 ijerph-17-09442-f006:**
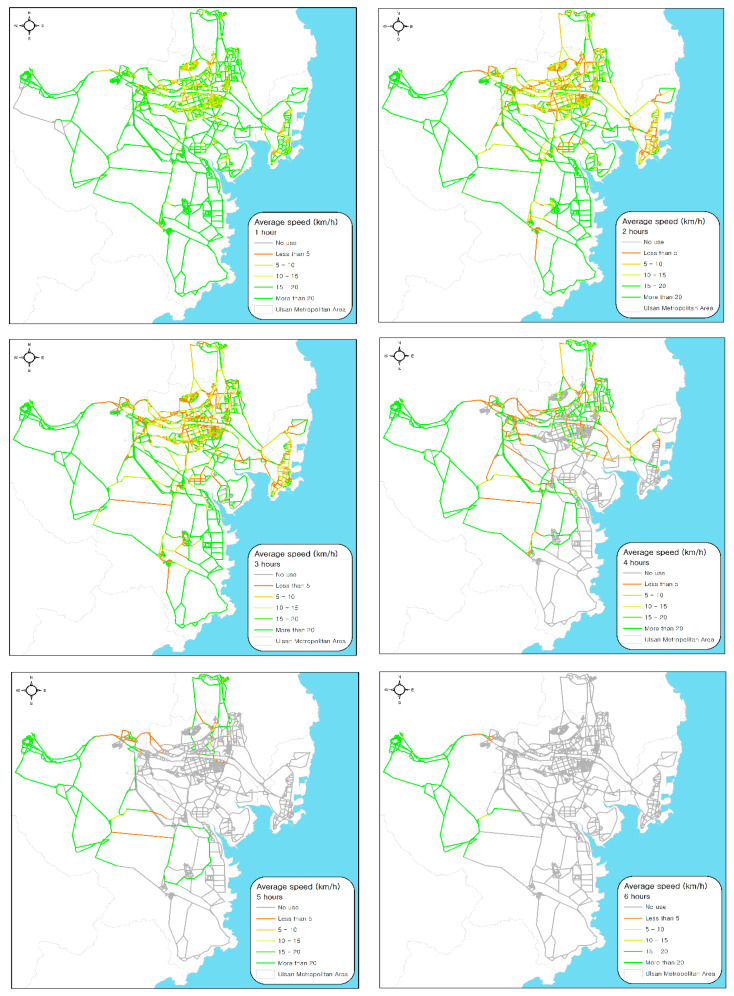
Average speed of vehicles from 1 to 6 h after the occurrence of accidents.

**Figure 7 ijerph-17-09442-f007:**
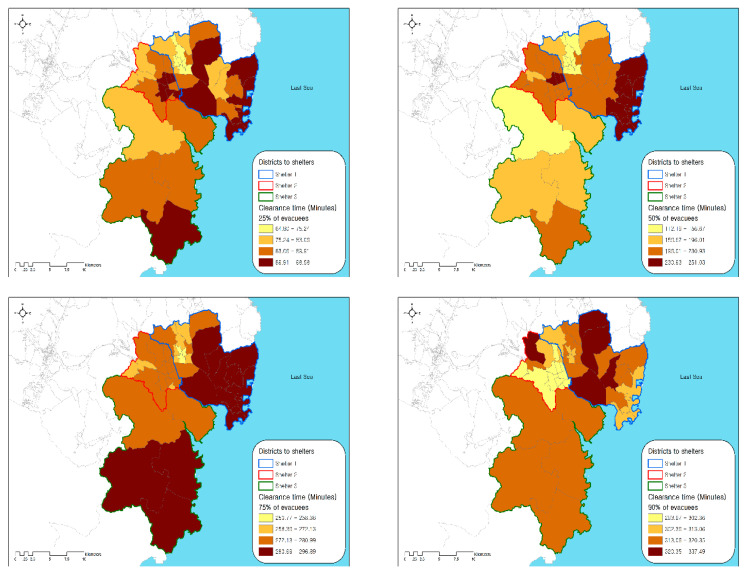
Clearance time required for 25%, 50%, 75%, and 90% of evacuation.

**Figure 8 ijerph-17-09442-f008:**
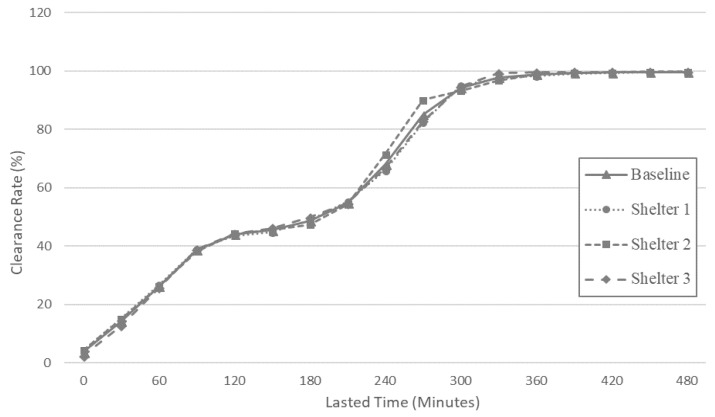
Clearance rate of the baseline scenario.

**Figure 9 ijerph-17-09442-f009:**
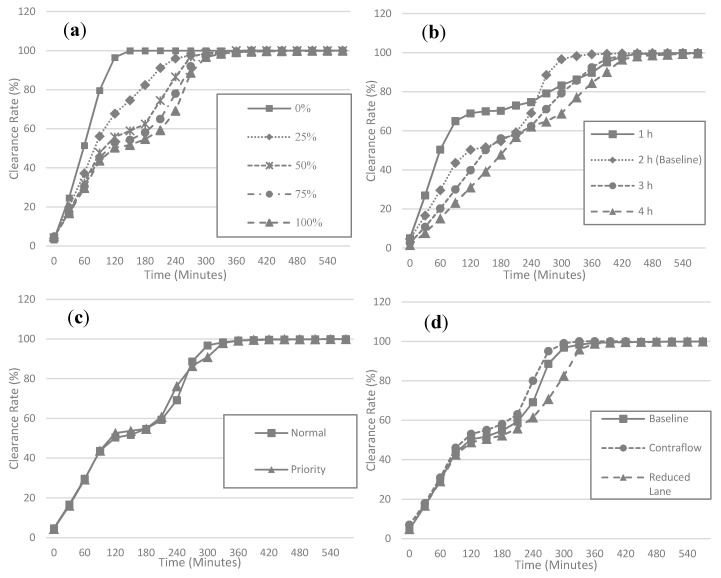
Sensitivity of (**a**) the proportion of private vehicles, (**b**) evacuation departure time period, (**c**) phasing policies, and (**d**) lane policies to clearance rate (%).

**Table 1 ijerph-17-09442-t001:** Summary of ALOHA input data used in the study.

Input Data		Value
Local atmospheric conditions	Wind speed (m/s)	1.5
	Wind direction	No direction
	Measurement height (m)	10
	Ground roughness	Urban and forest
	Cloud cover	Partly cloudy
	Air temperature (°C)	25
	Humidity (%)	50
Identity of chemical	Type of chemical	19 toxic chemicals
	Status of chemical	Pure chemicals
Details about spill scenario	Source type	Direct
	Leakage rate (kg/min)	Stored chemicals/60 min
	Height of chemical release	Ground level

**Table 2 ijerph-17-09442-t002:** Required information for network input files.

Network Component	Column	Description
Node	NODE	Node ID number
	X_COORD	X coordinate of node
	Y_COORD	Y coordinate of node
Link	ID	Link ID number
	ANODE	ID number of node at one end of the link
	BNODE	ID number of node at the other end of the link
	LANES_BA	Number of lanes in the direction from B to A
	LANES_AB	Number of lanes in the direction from A to B
	LENGTH	Length of the link in meters
	TYPE	Functional class of the road
	USE	Vehicle types that are allowed to use the link
ZONE	ID	Zone ID number
	X_COORD	X coordinate of zone centroid
	Y_COORD	Y coordinate of zone centroid

**Table 3 ijerph-17-09442-t003:** Estimated radius of 19 toxic chemicals’ release in each ERPG level.

Toxic Chemical	ERPG-2 Level		ERPG-3 Level	
	Radius (km)	Concentration (ppm)	Radius (km)	Concentration (ppm)
Formaldehyde	7.84	2.6	3.92	11
Benzene	0.46	150	0.17	1000
Methyl chloride	0.21	1000	0.12	3000
Hydrogen cyanide	2.38	10	1.29	25
Vinyl chloride	0.46	5000	0.18	20,000
Ethylene oxide	1.35	50	0.35	500
Propylene oxide	0.41	250	0.22	750
Methyl ethyl ketone	0.6	2700	0.5	4000
Toluene	1.59	300	0.89	1000
Ethyl acetate	0.18	1700	0.08	10,000
Hydrogen chloride	7.7	20	3.85	150
Hydrogen fluoride	1.08	50	0.84	20
Ammonia	2.91	150	0.65	1500
Sulfuric acid	5.4	7.3	2.7	130
Nitric acid	6.05	10	4.78	78
Phosphorus trichloride	3.35	3	1.19	15
Chlorine	5.32	3	1.62	20
Hydrogen sulfide	0.83	30	0.32	100
Chlorosulfonic acid	5	10	2.52	30

**Table 4 ijerph-17-09442-t004:** Average speed of vehicles from 1 to 6 h after the occurrence of accidents.

Average Speed (km/h)		<1 h	<2 h	<3 h	<4 h	<5 h	<6 h	Total Average
No use	# of segments	339	130	535	3606	5115	5687	40
	% of segments	5.48	2.10	8.65	58.30	82.70	91.95	0.65
Less than 5	# of segments	557	1420	1587	570	152	24	2063
	% of segments	9.01	22.96	25.66	9.22	2.46	0.39	33.35
5 to 10	# of segments	1112	1233	905	285	73	17	2118
	% of segments	17.98	19.94	14.63	4.61	1.18	0.27	34.24
10 to 15	# of segments	1221	1206	965	330	111	30	1224
	% of segments	19.74	19.50	15.60	5.34	1.79	0.49	19.79
15 to 20	# of segments	941	681	640	290	127	43	303
	% of segments	15.21	11.01	10.35	4.69	2.05	0.70	4.90
More than 20	# of segments	2015	1515	1553	1104	607	384	437
	% of segments	32.58	24.49	25.11	17.85	9.81	6.21	7.07

**Table 5 ijerph-17-09442-t005:** Sensitivity of factors to clearance time.

Factors		25% of Evacuation (Minutes)	50% of Evacuation (Minutes)	75% of Evacuation (Minutes)	90% of Evacuation (Minutes)
Proportion of private vehicles	100% (Baseline)	49.31	118.51	249.04	275.13
	75%	47.42	108.66	233.08	265.71
	50%	44.97	98.47	211.38	250.08
	25%	38.80	80.24	151.65	205.94
	0%	30.48	58.42	85.12	108.45
Departure time range	1 h	27.27	59.49	244.78	361.57
	2 h (Baseline)	49.31	118.51	249.04	275.13
	3 h	74.60	149.26	284.28	348.91
	4 h	96.68	187.10	322.10	389.07
Phasing policy	Normal (Baseline)	49.31	118.51	249.04	275.13
	Phased	50.28	111.18	237.65	294.00
Lane Policy	Normal (Baseline)	49.31	118.51	249.04	275.13
	Contraflow	46.15	107.14	231.18	260.00
	Reduced Lane	50.05	141.08	280.85	316.77

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
