# Peer review of "Influence of Evacuation Policy on Clearance Time under Large-Scale Chemical Accident: An Agent-Based Modeling"

_ijerph, 2020, doi:10.3390/ijerph17249442_

Round 1
Reviewer 1 Report
The topic is relevant and the focus original as few studies have been already conducted on the context of chemical accidents. The agent-based evacuation model appears appropriate when combining the two approaches chosen. I really appreciates that the study have incorporated an estimation of the spatial extent of chemical accidents into evacuation modelling. Estimating reasonable spatial extent and evacuation demands can increase the preparedness respect the disaster scenarios. The methodology is correct and the results well explained. Nevertheless, the results are limited to a simple mobility scenario, not allowing to the needed complex simulation. The literature is slightly dated and lack in the resilience approaches that could useful to overcome the limits of the simulation tool.
Author Response
The topic is relevant and the focus original as few studies have been already conducted on the context of chemical accidents. The agent-based evacuation model appears appropriate when combining the two approaches chosen. I really appreciates that the study have incorporated an estimation of the spatial extent of chemical accidents into evacuation modelling. Estimating reasonable spatial extent and evacuation demands can increase the preparedness respect the disaster scenarios. The methodology is correct and the results well explained. Nevertheless, the results are limited to a simple mobility scenario, not allowing to the needed complex simulation. The literature is slightly dated and lack in the resilience approaches that could useful to overcome the limits of the simulation tool.
- We agree that this paper’s proposed methodology is limited to a simple evacuation scenario that may not fully account for the complexity of actual cases. However, we intended to examine the effects of the policy interventions on evacuation efficiency in a worst-case scenario. Although we cannot fully simulate the complexity of reality, we suppose that our finding may provide meaningful policy implications by comparing the influence of several interventions. We described these limitations in the discussion part (Page 14 line 364-366).
- We updated recent studies regarding the evacuation modeling as well as its limitations for chemical accident in this paper (Page 2 line 57-61, Page 13 line 358-359).
Reviewer 2 Report
The article is well written and interesting in terms of the results of various factors on clearance rates. Unique in merging chemical transport and evacuation models.
I have attached a file that provides some grammar improvement suggestions. Also included are a few questions and suggestions for figure changes.

Author Response
The article is well written and interesting in terms of the results of various factors on clearance rates. Unique in merging chemical transport and evacuation models.
I have attached a file that provides some grammar improvement suggestions. Also included are a few questions and suggestions for figure change
- We revised our manuscripts and reflected all of minor comments.
- Page 2 line 74, we described the location of ‘staging sites’ in Page 7 line 195-196 and Figure 5. We added parenthesis to link this.
- Page 6 line 167-168, we added the statement that ‘each chemical released will have its unique ERPG threat zones.
- Page 7 line 229-233, we described how the ‘evacuation phase control’ is defined in this paper.
- Page 9 line 259, we defined the baseline scenario in Figure 2, so we added parenthesis to link this.
- Figure 9, we revised the figure to show all legends
- Table 5, the ‘0% clearance time’ in Figure 9 corresponds with when the proportion of private vehicles are 0% (highlighted 5th row in the table). In column, 0% evacuation time would be 0.
Reviewer 3 Report
The authors discussed an interesting topic. Today's society is a high-risk society. The authors discussed the impact of evacuation policy on clearance time under large-scale chemical accident. This study help enhance people’s ability to respond to emergencies. Therefore, I suggest publishing the revised paper after minor revision. The revised suggestion is as follows:
The author discussed the impact of evacuation policies on clearance time, but I am more concerned about how to improve these policies. Therefore, I suggest that the authors should add more discussions on improving policies to help shorten the clearance time.
Author Response
The authors discussed an interesting topic. Today's society is a high-risk society. The authors discussed the impact of evacuation policy on clearance time under large-scale chemical accident. This study help enhance people’s ability to respond to emergencies. Therefore, I suggest publishing the revised paper after minor revision. The revised suggestion is as follows:
The author discussed the impact of evacuation policies on clearance time, but I am more concerned about how to improve these policies. Therefore, I suggest that the authors should add more discussions on improving policies to help shorten the clearance time.
- To emphasize the improving policies, we added discussions on several strategies that may increase the overall efficiency of evacuation (Page 13 line 355-363). First, we suggest the necessity of restricting private vehicle use during evacuation, instead utilizing public transit to reduce clearance time. Second, we suggest that mediating evacuation time periods can reduce overall clearance time since it disperses evacuation demands. Practitioners in evacuation planning should establish sequential evacuation plans and encourage people to be educated and trained in advance.
Reviewer 4 Report
This study focuses on the influence of evacuation policy on clearance time under large-scale chemical accident by means of an agent-based modeling. The paper addresses an important topic. However, it requires some work before it can be considered for publication. Some weak points have to be strengthened. Below please find more specific comments:
*Page 1 line 16: “TRANSIMS agent-based evacuation model with ALOHA” – what is ALOHA? Please specify.
*Page 1 line 21: “within 5 hours in the baseline scenario” – please be more specific regarding the baseline scenario.
*Pages 1-2: One of the major drawbacks of this work is lack of a solid literature review. There are a lot of studies that proposed different methods for emergency management (not just chemical accidents but also other hazards, e.g. hurricanes, fire). I suggest discussing some of the most relevant studies, including the following:
- Hou, J., Gai, W.M., Cheng, W.Y. and Deng, Y.F., 2020. Survey-based analysis of evacuation preparation behaviors in a chemical leakage accident: A case study. Journal of Loss Prevention in the Process Industries, 68, p.104219.
- Wang, J., Yu, X. and Zong, R., 2020. A dynamic approach for evaluating the consequences of toxic gas dispersion in the chemical plants using CFD and evacuation modelling. Journal of Loss Prevention in the Process Industries, p.104156.
- Dulebenets, M. A., Abioye, O. F., Ozguven, E. E., Moses, R., Boot, W. R., & Sando, T. (2019). Development of statistical models for improving efficiency of emergency evacuation in areas with vulnerable population. Reliability Engineering & System Safety, 182, 233-249.
- Li, Z., Huang, H., Li, N., Zan, M.L.C. and Law, K., 2020. An agent-based simulator for indoor crowd evacuation considering fire impacts. Automation in Construction, 120, p.103395.
- Abioye, O.F., Dulebenets, M.A., Ozguven, E.E., Moses, R., Boot, W.R. and Sando, T., 2020. Assessing perceived driving difficulties under emergency evacuation for vulnerable population groups. Socio-Economic Planning Sciences, p.100878.
- Hou, J., Gai, W.M., Cheng, W.Y. and Deng, Y.F., 2020. Statistical analysis of evacuation warning diffusion in major chemical accidents based on real evacuation cases. Process Safety and Environmental Protection.
Review and acknowledgment of the previous studies is critical for this paper to be solid.
*Page 2: Please provide a discussion that justifies selection of the study area.
*Page 4: Please provide a discussion that justifies selection of the ALOHA model.
*Page 13: “3.3. Sensitivity Testing” does not sound well. I suggest using “3.3. Sensitivity Analysis”.
*Pages 15-16: I suggest having two separate sections. One should be devoted to the discussion, and the last one will be devoted to conclusions.
Author Response
This study focuses on the influence of evacuation policy on clearance time under large-scale chemical accident by means of an agent-based modeling. The paper addresses an important topic. However, it requires some work before it can be considered for publication. Some weak points have to be strengthened. Below please find more specific comments:
*Page 1 line 16: “TRANSIMS agent-based evacuation model with ALOHA” – what is ALOHA? Please specify.
- We specified the term TRANSIMS and ALOHA in the abstract to clarify the term. Introduction of those two models are stated in Page 4 line 126-133, and Page 6 line 170-177.
*Page 1 line 21: “within 5 hours in the baseline scenario” – please be more specific regarding the baseline scenario.
- We specified the baseline scenario in the abstract to clarify the term. The description of the baseline scenario are shown in Figure 2.
*Pages 1-2: One of the major drawbacks of this work is lack of a solid literature review. There are a lot of studies that proposed different methods for emergency management (not just chemical accidents but also other hazards, e.g. hurricanes, fire). I suggest discussing some of the most relevant studies, including the following:
- Hou, J., Gai, W.M., Cheng, W.Y. and Deng, Y.F., 2020. Survey-based analysis of evacuation preparation behaviors in a chemical leakage accident: A case study. Journal of Loss Prevention in the Process Industries, 68, p.104219.
- Wang, J., Yu, X. and Zong, R., 2020. A dynamic approach for evaluating the consequences of toxic gas dispersion in the chemical plants using CFD and evacuation modelling. Journal of Loss Prevention in the Process Industries, p.104156.
- Dulebenets, M. A., Abioye, O. F., Ozguven, E. E., Moses, R., Boot, W. R., & Sando, T. (2019). Development of statistical models for improving efficiency of emergency evacuation in areas with vulnerable population. Reliability Engineering & System Safety, 182, 233-249.
- Li, Z., Huang, H., Li, N., Zan, M.L.C. and Law, K., 2020. An agent-based simulator for indoor crowd evacuation considering fire impacts. Automation in Construction, 120, p.103395.
- Abioye, O.F., Dulebenets, M.A., Ozguven, E.E., Moses, R., Boot, W.R. and Sando, T., 2020. Assessing perceived driving difficulties under emergency evacuation for vulnerable population groups. Socio-Economic Planning Sciences, p.100878.
- Hou, J., Gai, W.M., Cheng, W.Y. and Deng, Y.F., 2020. Statistical analysis of evacuation warning diffusion in major chemical accidents based on real evacuation cases. Process Safety and Environmental Protection.
Review and acknowledgment of the previous studies is critical for this paper to be solid.
- We reviewed suggested studies and added relevant studies in this paper. First, we updated the literature review with some of relevant studies (Page 2 line 57-61). Here, we revised the drawbacks of previous studies are absent of simulating large-scale chemical accidents and corresponding mass evacuation. Second, we updated the discussion part (Page 13 line 358-359) with recent studies.
*Page 2: Please provide a discussion that justifies selection of the study area.
- We described the reasons for choosing the study area in page 3 line 95-102. The study area is one of the most vulnerable regions to chemical accidents since it has large industrial complexes over 40 years, and there are largely-populated regions nearby those complexes.
*Page 4: Please provide a discussion that justifies selection of the ALOHA model.
- We described why the ALOHA model was selected in page 4 line 126-133. Although ALHA model is not exquisite as other dispersion models, this model is more intuitive to use, and findings are easy to understand.
*Page 13: “3.3. Sensitivity Testing” does not sound well. I suggest using “3.3. Sensitivity Analysis”.
- We changed the term into “Sensitivity Analysis” (Page 12 line 310)
*Pages 15-16: I suggest having two separate sections. One should be devoted to the discussion, and the last one will be devoted to conclusions.
- We separated the discussion and conclusion parts (Page 13-14). In discussion part, we organized the findings of the study and policy implications to improve efficiency of evacuation. Also, we states several limitations for this study. In conclusion part, we emphasized the drawbacks of previous works and the strength of our study.
Round 2
Reviewer 4 Report
The authors made some improvements in the original version of the paper. However, there are still some areas for improvement before the manuscript can be considered for publication:
*Page 1 line 22: “Evacuating maximum number of private vehicles” should be replaced with “evacuating maximum number of private vehicles” as there is no point to capitalize “evacuating” inside the parenthesis.
*I still feel that my previous comment regarding the literature review part has not been adequately addressed. The authors just added a couple references, which still does not seem convincing. Plus, I see that the authors mostly acknowledge the studies that focused on agent-based methodologies, which may seem limited to some readers. At the beginning of the introduction section, I recommend adding a broader discussion regarding different methods that were proposed for emergency management (not just chemical accidents but also other hazards, e.g. hurricanes, fire) and not just agent-based methodologies but also optimization methods, driving simulation methods, machine learning methods, and other advanced analytical methods. This discussion should be supported by the relevant references, including the following:
- Tian, Z., Zhang, G., Hu, C., Lu, D. and Liu, H., 2020. Knowledge and emotion dual-driven method for crowd evacuation. Knowledge-Based Systems, 208, p.106451.
- Wang, J., Yu, X. and Zong, R., 2020. A dynamic approach for evaluating the consequences of toxic gas dispersion in the chemical plants using CFD and evacuation modelling. Journal of Loss Prevention in the Process Industries, p.104156.
- Dulebenets, M. A., Abioye, O. F., Ozguven, E. E., Moses, R., Boot, W. R., & Sando, T. (2019). Development of statistical models for improving efficiency of emergency evacuation in areas with vulnerable population. Reliability Engineering & System Safety, 182, 233-249.
- Wang, K., Shi, X., Goh, A.P.X. and Qian, S., 2019. A machine learning based study on pedestrian movement dynamics under emergency evacuation. Fire safety journal, 106, pp.163-176.
- Abioye, O.F., Dulebenets, M.A., Ozguven, E.E., Moses, R., Boot, W.R. and Sando, T., 2020. Assessing perceived driving difficulties under emergency evacuation for vulnerable population groups. Socio-Economic Planning Sciences, p.100878.
- Hou, J., Gai, W.M., Cheng, W.Y. and Deng, Y.F., 2020. Statistical analysis of evacuation warning diffusion in major chemical accidents based on real evacuation cases. Process Safety and Environmental Protection.
This will make the literature review part more reasonable and thorough.
*Page 2 line 78: It may seem inappropriate for some readers when the author refer to Fig. 5 before referring to Fig. 1. Please consider revising.
*Page 3 line 100: “within a 10km radius” should be replaced with “within a 10 km radius”.
*Page 4 line 132: “We employed version 5.4.7 (published in September 2016)” – the term “published” does not fit well here. I suggest using “released in September 2016”.
*Page 14: I see that the authors have a conclusion section now. However, it seems pretty short. This section should be strengthened. The authors should expand on the limitations of this study and how they will be addressed in future research.
Author Response
Please find an attached document
